# IL-8, TNF-α, and IL-17 in the Development of Erosive Esophagitis and Symptom Perception in Gastroesophageal Reflux Disease (GERD)

**DOI:** 10.3390/jcm13195832

**Published:** 2024-09-29

**Authors:** Titong Sugihartono, Amal Arifi Hidayat, Ricky Indra Alfaray, Michael Austin Pradipta Lusida, Isna Mahmudah, Hafeza Aftab, Ratha-Korn Vilaichone, Yoshio Yamaoka, Hoda M. Malaty, Muhammad Miftahussurur

**Affiliations:** 1Division of Gastroenterology-Hepatology, Department of Internal Medicine, Faculty of Medicine, Universitas Airlangga, Surabaya 60132, Indonesia; titongsppd@gmail.com; 2Department of Internal Medicine, Faculty of Medicine, Universitas Airlangga, Surabaya 60132, Indonesia; arifiamal@gmail.com (A.A.H.); michaellusida@gmail.com (M.A.P.L.); isnamahmudah@gmail.com (I.M.); 3Department of Environmental and Preventive Medicine, Faculty of Medicine, Oita University, Yufu 879-5593, Oita, Japan; hmalaty@bcm.tmc.edu (R.I.A.); yyamaoka@oita-u.ac.jp (Y.Y.); 4Helicobacter Pylori and Microbiota Study Group, Institute Tropical Disease, Universitas Airlangga, Surabaya 60115, Indonesia; 5Department of Gastroenterology, Dhaka Medical College and Hospital, Dhaka 1000, Bangladesh; rosefebruary28@yahoo.com; 6Center of Excellence in Digestive Diseases and Gastroenterology Unit, Department of Medicine, Faculty of Medicine, Thammasat University, Pathum Thani 12120, Thailand; 7Chulabhorn International College of Medicine (CICM), Thammasat University, Pathum Thani 12120, Thailand; 8Department of Medicine, Baylor College of Medicine, Houston, TX 77030, USA

**Keywords:** proinflammatory cytokines, erosive esophagitis, GERD, human and health

## Abstract

**Background:** The diverse clinical characteristics of erosive esophagitis (EE) and symptom perception in patients with gastroesophageal reflux disease (GERD) remain a major challenge in understanding their underlying pathogenesis. This study aimed to investigate the association between the levels of IL-8, TNF-α, and IL-17 in serum and the presence of erosive esophagitis and symptoms related to GERD. **Method:** We enrolled 65 subjects presenting with GERD symptoms. Based on the findings of upper endoscopy, the subjects were categorized into two groups: (1) erosive esophagitis (EE LA grades B-D) and (2) non-erosive esophagitis (normal-EE LA grade A). Symptom perception was assessed via GERD questionnaire (GERD-Q) and the frequency scale for the symptoms of GERD (FSSG). The enzyme-linked immunosorbent assay (ELISA) method was used to analyze serum levels of IL-8, TNF-α, and IL-17. Analysis of cytokine levels between different symptoms severity was performed using the Kruskal-Wallis H test. **Results:** Median serum IL-8 levels were significantly higher in the erosive esophagitis group compared to those with non-erosive esophagitis (20.2 (IQR 16.9–32.2) vs. 17.7 (IQR 15.2–19.6), *p* < 0.05). The study found a significant association between IL-8 levels and the presence of globus symptoms (median IL8 level 46.961 (38.622–92.644) in subjects with globus vs. 18.06 (16.68–20.49) in those without globus; *p* < 0.05). Similarly, TNF-α levels were associated with the frequency of regurgitation symptoms (H index = 10.748; dr = 3; *p* < 0.05). We observed a significant correlation between IL-17 levels and the frequency of heartburn and early satiety symptoms. **Conclusions:** IL-8 may play a role in the development of mucosal erosion in GERD. IL-8, TNF- α, and IL-17 might be involved in the development of globus symptoms, the frequency of regurgitation, and the frequency of heartburn and early satiety, respectively. The diverse symptom phenotypes observed in patients with GERD symptoms may be mediated by distinct profiles of proinflammatory cytokines.

## 1. Introduction

GERD is a disorder characterized by the repeated reflux of gastric contents into the esophageal lumen, resulting in troublesome symptoms and/or complications [1]. The global prevalence of GERD is estimated to be 13.9% in the general population [2], with a substantial rise of 77.5% over the past three decades [3]. Despite the fact that the prevalence of GERD in Asian populations is lower than the global average (2.5–7.8%) [4], a number of studies demonstrate that its incidence continues to rise annually [4,5,6]. GERD has been widely recognized as an illness that may substantially reduce quality of life [7]. GERD patients can be classified into two groups based on endoscopy findings: erosive esophagitis (EE), characterized by erosion of the esophageal mucosa, and non-erosive reflux disease (NERD), which lacks evidence of esophageal mucosal damage [8].

One of the most challenging aspects of managing patients with GERD is accommodating their vast range of diverse clinical characteristics. Only 30–40% of those with GERD symptoms are found to have evidence of mucosal lesions (EE and/or Barrett’s esophagus), whereas the remaining 60–70% turn out to have no visible mucosal damage [9]. In one study, neither the severity of reflux measured by pH monitoring nor the mucosal erosion observed by endoscopy was correlated with GERD symptoms [10]. Despite being a longstanding and effective treatment for GERD, proton pump inhibitors (PPIs) also fail to deliver an adequate response in about 30% of patients [11].

The etiology of esophageal mucosal damage in GERD has traditionally been attributed to caustic injury resulting from direct acid exposure [12]. Recently, however, a study has emerged to challenge this paradigm by demonstrating that immune cell infiltration precedes the presence of mucosal erosions [13]. Acid exposure can stimulate mucosal cells to produce proinflammatory cytokines, thereby initiating the migration of immune cells [13]. The potential roles of pro-inflammatory cytokines in GERD have been investigated in previous studies [13,14,15]. However, there are not many studies examining the relationship between the cytokines IL-8, TNF-α, and IL-17 and GERD. This study aims to investigate the association between the levels of IL-8, TNF-α, and IL-17 in serum and the presence of erosive esophagitis, as well as the perception of symptoms related to GERD.

## 2. Methods

### 2.1. Study Design and Population

This was a cross-sectional study conducted at the gastroenterology polyclinics of two hospitals: RSUD Dr. Soetomo Surabaya, Indonesia, and Siti Khadijah Hospital Sidoarjo, Indonesia. We enrolled adult patients (≥18 years old) who had symptoms of heartburn and/or regurgitation at least twice a week for a duration of three months or longer. We performed an esophagogastroduodenoscopy (EGD) on all subjects, followed by a mucosal biopsy sampling of the distal third of the esophagus. Subjects were instructed not to take proton pump inhibitors (PPIs) for the two weeks prior to EGD. The use of antacids was permitted for symptom relief during these periods. The exclusion criteria for this study were a history of autoimmune disease, cirrhosis, cancer, pregnancy, signs of infection, and findings of endoscopic pathology other than erosive esophagitis or Barrett’s esophagus. All subjects provided informed consent before participating in this study. The study protocol was approved by the ethics committee of RSUD Dr. Soetomo Surabaya (0124/KPEK/1/2021).

### 2.2. Data Collection

The study collected demographic and clinical data, including age, gender, ethnicity, and body mass index (BMI). The upper endoscopy examination was performed by a Gastroenterologist with over 5 years of clinical experience. We used the Los Angeles (LA) classification system to describe the severity of erosive esophagitis (EE): LA Grade A, one (or more) mucosal breaks ≤ 5 mm that do not extend between the tops of two mucosal folds; LA grade B, one (or more) mucosal breaks > 5 mm that do not extend between the tops of two mucosal folds; LA grade C, one (or more) mucosal breaks that are continuous between the tops of two or more mucosal folds but involve <75% of the circumference; LA grade D, one (or more) mucosal breaks that involve ≥75% of the circumference [9]. According to the 2018 Lyon consensus, LA grades B–D mucosal erosion can be used to objectively diagnose erosive esophagitis. LA grade A esophagitis has been found in many asymptomatic healthy individuals and is therefore not conclusive for GERD [16]. In this study, we classified subjects with LA grades B–D as the erosive esophagitis (EE) group, while those with normal mucosa and LA grade A were classified as the non-erosive (non-EE) group. Hematoxylin-eosin staining was used for histopathological examination of the esophageal tissue biopsy. All subjects completed the GERD questionnaire (GERD-Q) and the frequency scale for the symptoms of GERD (FSSG) to quantify their symptom perception. These two questionnaires had been translated into and Bahasa Indonesia and validated for use in previous studies [17,18].

Blood samples were collected prior to the EGD using a serum separator tube (SST) container (BD Vacutainer^®^). Following the centrifugation process, the serum was stored in a cryotube at a temperature of −80 °C until analysis. IL-8, TNF-α, and IL-17 levels were measured by the enzyme-linked immunosorbent assay method using Quantikine Human Immunoassays (R&D systems Inc., Minneapolis, MN, USA) [19]. The plates were read and analyzed via Microplate Reader Biorad model 680 (Bio-rad laboratories Inc., Hercules, CA, USA). Measurements were carried out in the 450 nm wavelength.

### 2.3. Statistical Analysis

Categorical variables are presented by frequency (n) and percentage (%), while continuous variables are expressed as mean (standard deviation) or median (interquartile range). Univariate analysis was performed on demographic variables (sex, age, and ethnicity) and body mass index (BMI) to assess the differences between the erosive and non-erosive groups. Non-parametric analysis was applied in this study due to the abnormal distribution of IL-8, TNF-α, and IL-17 levels. The levels of these three cytokines were also compared in the erosive and non-erosive esophagitis groups (EE vs. non-EE). Additionally, a sub-analysis was performed to compare the levels of these cytokines at each degree of mucosal erosion, as classified originally by the Los Angeles system (normal, LA grade A, B, C, D). The correlation between these three cytokine levels and the total scores of GERDQ and FSSG was evaluated. We also evaluated the association between each cytokine and the frequency/presence of various symptoms presented in the questionnaires, including heartburn, regurgitation, nausea, sleep disturbance, bloating, post-prandial fullness, early satiety, belching, globus, odynophagia, dysphagia, hoarseness, post-nasal drip, dyspnea, and cough. In order to detect the difference in cytokine levels among different degrees of symptom severity, we used the Kruskal-Wallis test. All statistical analyses were performed using the IBM SPSS version 25 (IBM Co., Armonk, NY, USA).

## 3. Results

### 3.1. Clinical Characteristics

Out of 82 eligible subjects, 2 declined participation, and 15 were excluded due to the finding of other pathologies upon endoscopy (10 cases of erosive gastritis, 1 case of gastric polyp, and 4 cases of gastric ulcers). Esophageal biopsy was taken from 64 of the subjects included in this study, 28 of which were excluded from analysis due to sample defects. The study population was predominately female (63%)—with an average age of 40.4 ± 14.2 years and a mean BMI of 23.9 ± 7.8 kg/m^2^—and was ethnically diverse; 72.3% were Javanese, 16.9% were Madurese, 4.6% were Sundanese, and 6.1% were Chinese (Table 1).

The upper endoscopy examination showed that 51 subjects (78.5%) had erosive esophagitis, with 19 (29.3%) of them classified as LA grades B and C. Most subjects (70.7%) demonstrated no mucosal damage or were merely classified as LA grade A. Neither LA grade D erosive esophagitis nor Barrett’s esophagus were identified in the studied population. Based on symptom perception, the study population had a median GERD-Q score of 9 (8–12) and FSSG score of 19 (15–46). The median levels of IL-8, TNF-α, and IL-17 in all subjects were 18.3 (15.9–21.0) ng/L, 91.3 (66.4–143.0) ng/L, and 48.2 (39.5–116.1) ng/L, respectively. Histopathology analysis revealed that 6.7% of subjects had chronic esophagitis, 11.1% had glycogenic acanthosis, and 72.2% had normal results. There were no statistically significant differences in terms of gender, age, and ethnicity between the erosive and non-erosive groups.

#### 3.1.1. The Association between Serum Levels of IL-8, TNF-α, and IL-17 and the Severity of Mucosal Erosion and Histopathological Features of the Esophagus

There were significantly elevated levels of IL-8 in the erosive esophagitis group compared to the non-erosive group (*p* < 0.05). Table 2 summarizes the comparison of IL-8, TNF-α, and IL-17 serum levels between patients with significant erosive esophagitis (LA grades B–C) and those without such evidence (normal–LA grade A). Figure 1 displays a comparative graph illustrating the serum IL-8 concentrations between the erosive and non-erosive groups. There was no statistically significant difference in the levels of TNF-α and IL-17 between the two groups (*p* > 0.05). Based on the original Los Angeles classification, a sub-analysis that compared the levels of these three cytokines at each level of mucosal erosion also showed that IL-8 is significantly different (*p* < 0.05). However, there was no statistically significant association observed between the levels of the three cytokines and the histopathological features of the esophageal tissue biopsies (*p* > 0.05).

#### 3.1.2. The Association between Serum Levels of IL-8, TNF-α, and IL-17 and Symptom Perception

There were no statistically significant differences when IL-8, TNF-, and IL-17 serum levels were compared to the total scores of both GERD-Q and FSSG (*p* > 0.05). The study found significant correlations between these cytokines and four different symptom phenotypes. Certain serum levels of IL-8 were found to be associated with the presence of globus symptoms (*p* < 0.05), whereas the presence of TNF-α exhibited a significant correlation with both the frequency and presence of regurgitation symptoms (*p* < 0.05). On the other hand, IL-17 was discovered to be associated with a broader range of symptoms in GERD, including both the frequency and presence of heartburn, as well as the presence of early satiety. Details about the relationship between the cytokine levels and the presence of the symptoms are provided in Table 3. Table 4 provides the details of correlations between the frequency or severity of symptoms and cytokine levels.

## 4. Discussion

Our study observed a significant rise in IL-8 serum levels among patients experiencing heartburn and/or regurgitation symptoms, who also had objective evidence of GERD in the form of erosive esophagitis, compared to those lacking such evidence. This finding is in line with a previous study that has reported comparable results [20]. This study observed that patients who did not respond to standard PPI therapy demonstrated higher levels of IL-8 compared to those who had an adequate response. However, the study reported that serum IL-8 showed no correlation with the severity of mucosal erosion [20]. High IL-8 serum levels have been linked to recurrent erosions, even among patients who have received standard therapy [21]. An investigation that incorporated pH monitoring documented a correlation between high serum IL-8 levels and an increase in both the number of occurrences of acid reflux and acid exposure time (AET) [21]. IL-8 is a highly potent chemotactic factor for neutrophils and T lymphocytes [22,23]. Furthermore, this cytokine plays a role in the activation of neutrophils, leading to the development of localized tissue damage. An in vitro-based study using a culture of esophageal mucosa demonstrated that acid exposure (caustic damage) did not directly induce cell death. However, this state promotes epithelial cell secretion of IL-8, which can lead to the migration of inflammatory cells into the local tissue [13]. This event is driven by the generation of reactive oxygen species (ROS), followed by the activation of hypoxia-inducible factor (HIF)-2α [24]. Our study demonstrates that a rise in serum levels may suggest a significant production of IL-8 by the esophageal epithelium. The correlation between IL-8 and the formation of mucosal erosion indicates its significant involvement.

This study also revealed significant correlations between the serum levels of IL-8, TNF-α, and IL-17 and many different symptom perceptions. This phenomenon is a novel discovery that has not been previously documented. Heartburn and regurgitation are both typical symptoms of GERD; however, their underlying mechanisms differ substantially [25]. Heartburn is a symptom characterized by a sensation of burning in the substernal area, which may radiate towards the neck [25]. Acid exposure may provoke heartburn symptoms by activating an acid-sensitive receptor located on afferent nerve endings within the esophageal squamous epithelial tissue. Several nociceptive receptors have been identified as being involved in this process, including the transient receptor potential vanilloid type-1 (TRPV1), protease-activated receptors (PAR2), and acid-sensing ion channels (ASICs) [26,27,28]. Direct activation and enhanced sensitivity of these receptors can be initiated by the release of proinflammatory cytokines in response to acid exposure [29,30]. Our findings suggest that IL-17 plays an important role in the underlying mechanism of heartburn symptoms. A recent genetic ontology study has identified the predominant biochemical pathways involved in GERD; the IL-17 signaling pathway was found to be dysregulated among the seven proinflammatory pathways studied [31]. Our study also observed a correlation between IL-17 and early satiety symptoms. This symptom has been reported to have a strong association with gastroparesis, as it falls within the spectrum of dyspepsia [32]. Cajal interstitial cell damage observed in gastroparesis was associated with a reduction in the number of M2 macrophages, which are anti-inflammatory in nature [33]. The results of this study provide evidence for the involvement of IL-17 in this process. Regurgitation refers to the effortless upward movement of stomach contents, typically accompanied by a sour or bitter taste [25]. These symptoms are attributed to transient relaxation of the upper and lower esophageal sphincter, decreased peristaltic amplitude, and large reflux volumes [34,35]. Our study revealed a correlation between serum TNF-α levels and regurgitation symptoms, suggesting the involvement of this cytokine in the development of this mechanical process. Globus is an extraesophageal symptom of GERD that manifests as the perception of a foreign object inside the throat [36]. The current understanding is that the mechanisms responsible for globus involve both direct activation of the vagus reflex and repeated acid reflux, leading to laryngeal mucosa damage [36]. Our findings suggest that IL-8 plays a significant role in the development of these symptoms. The study observed no correlation between serum levels of IL-17 and TNF-α and the presence of erosive esophagitis. Consistent with our findings, a research team discovered that there was no significant correlation between IL-17 serum levels and the presence of erosive esophagitis [37]. In contrast to asymptomatic healthy individuals, however, patients with erosive esophagitis tend to have higher IL-17 serum levels [38]. A recent study found no significant difference in TNF-α production between patients diagnosed with erosive esophagitis and non-erosive reflux disease (NERD). Nevertheless, patients with NERD, demonstrating abnormal acid exposure as determined by pH-monitoring studies, showed overexpression of TNF-α [39]. Although this study did not establish a correlation between IL-17 and TNF-α and the development of mucosal erosions, their involvement in the perception of specific symptoms suggests that these cytokines play a significant role in the pathogenesis of GERD symptoms.

While this research has generated several novel findings, we have identified certain limitations in the study. First, this study lacked control groups comprising healthy individuals. Second, since the sample is serum, the measured cytokines’ levels are not specific to the esophagus. Conditions such as autoimmune diseases, infections, or malignancies may also increase the cytokines’ levels. We have tried to limit this by excluding subjects with such conditions; however, we realize that this is limited. Further studies measuring local production of cytokine levels are required. Lastly, we did not include pH monitoring as a diagnostic method; hence, subjects in the non-erosive group could not be distinguished as having NERD, esophageal hypersensitivity, or functional heartburn.

## 5. Conclusions

IL-8 is a cytokine that is significantly involved in the development of erosive esophagitis in patients with typical GERD symptoms. The diverse symptom phenotypes observed in patients with GERD symptoms are determined by different proinflammatory cytokine profiles. IL-17 is predominantly associated with heartburn and early satiety symptoms. TNF-α is implicated in regurgitation symptoms, whereas IL-8 is associated with the presence of globus symptoms.

## Figures and Tables

**Figure 1 jcm-13-05832-f001:**
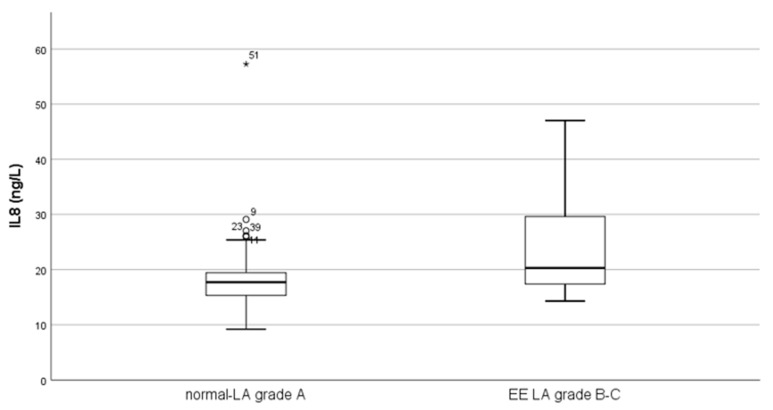
Comparison of IL-8 serum levels in the non-EE (normal–LA grade A) vs. EE (LA grades B–C) group. Abbreviations: LA grades A–D EE, Los Angeles grades A–D erosive esophagitis; ng/L, nanogram per liter; IL-8, interleukin-8.

**Table 1 jcm-13-05832-t001:** Clinical characteristics of subjects.

Characteristics (*n* = 65)	*n* (%) or Mean ± SD or Median (IQR)	*p*
Sex		>0.05
Male	25 (36.9)	
Female	41 (63.1)	
Age (years)	40.4 ± 14.2	>0.05
Body mass index (BMI) (kg/m^2^)	23.9 ± 7.8	>0.05
Ethnicity		>0.05
Javanese	47 (72.3)	
Madurese	11 (16.9)	
Sundanese	3 (4.6)	
Chinese	4 (6.1)	
EGD findings		
Normal	14 (21.5)	
LA grade A EE	32 (49.2)	
LA grade B EE	17 (26.2)	
LA grade C EE	2 (3.1)	
LA grade D EE	0 (0)	
Barrett’s esophagus	0 (0)	
GERDQ score	9 (8–12)	
IL-8 serum level (ng/L)	18.3 (15.9–21.0)	
TNF-α serum level (ng/L)	91.3 (66.4–143.0)	
IL-17 serum level (ng/L)	48.2 (39.5–116.1)	
Histopathology results (*n* = 36)		
Normal	26 (72.2)	
Chronic esophagitis	6 (16.7)	
Glycogenic acanthosis	4 (11.1)	

Abbreviations: SD, standard deviation; IQR, interquartile range; BMI, body mass index; LA grade A-D EE, Los Angeles grade A-D erosive esophagitis; GERD-Q, gastroesophageal reflux disease questionnaire; IL-8, interleukin-8; TNF-α, *tumor necrosis factor*-α; IL-17, interleukin-17.

**Table 2 jcm-13-05832-t002:** Comparison of IL-8, TNF-α, and IL-17 serum levels between the non-erosive esophagitis and erosive esophagitis groups.

Median Serum Levels (IQR)	Non-EE (Normal–LA Grade A)	EE (LA Grade B–C)	*p*
IL-8 (ng/L)	17.7 (15.2–19.6)	20.2 (16.9–32.2)	0.02
TNF-α (ng/L)	90.3 (65.3–172.4)	93.7 (72.9–147.8)	>0.05
IL-17 (ng/L)	53.2 (39.0–137.5)	47.9 (39.8–58.0)	>0.05

Abbreviations: LA grades A–D EE, Los Angeles grades A–D erosive esophagitis; ng/L, nanogram per liter; IL-8, interleukin-8; TNF-α, *tumor necrosis factor*-α; IL-17, interleukin-17.

**Table 3 jcm-13-05832-t003:** Correlation of IL-8, TNF-, and IL-17 serum levels with the presence of symptoms.

	Median		IL-8 (*p*)	Median		TNF-α (*p*)	Median		IL-17 (*p*)
	Present	Absent		Present	Absent		Present	Absent	
The presence of heartburn	17.715 (15.750–24.3375)	18.81 (17.54–20.315)	>0.05	90.556 (64.269–170.641)	91.347 (68.443–125.881)	>0.05	69 (46.137–156.66)	45.727 (35.886–60.659)	0.01 *
The presence of regurgitation	17.96 (15.698–20.49)	18.31 (16.895–23.810)	>0.05	92.541 (66.162–136.614)	85.058 (64.862–248.537)	0.04 *	49.467 (40.456–95.971)	48.207(32.215–125.549)	>0.05
The presence of epigastric pain	17.960 (16.1275–20.315)	18.81 (15.535–23.810)	>0.05	90.357 (69.999–362.768)	91.744 (59.640–125.743)	>0.05	47.791 (36.131–128.853)	53.74 (42.564–107.014)	>0.05
The presence of nausea	18.17 (15.75–21.095)	18.38 (16.79–23.215)	>0.05	87.795 (59.552–170.641)	100.607 (91.347–125.303)	>0.05	49.255 (38.917–128.853)	47.999 (42.275–62.013)	>0.05
The presence of sleep disturbance	17.68 (15.75–20.665)	18.985 (16.503–24.443))	>0.05	91.347 (61.772–131.177)	94.186 (70.928–250.736)	>0.05	45.583 (38.376–67.706)	71.280 (40.208–158.262)	>0.05
The presence of bloating	18.06 (15.68–20.35)	19.755 (16.59–28.49)	>0.05	91.347 (63.728–147.766)	95.919 (73.71–169.604)	>0.05	47.583 (39.065–153.443)	52.021 (40.254–89.445)	>0.05
The presence of post-prandial fullness	18.045 (15.698–21.001)	18.81 (16.5–21.22)	>0.05	91.347 (60.794–145.394)	93.738 (75.684–324.668)	>0.05	47.272 (38.129–114.403)	53.74 (47.48–122.697)	>0.05
The presence of early satiety	18.06 (17.54–21.197)	16.68 (14.88–21.18)	>0.05	88.777 (62.75–140.651)	99.792 (91.347–529.467)	>0.05	69.406 (53.74–159.869)	46.961 (38.622–92.6435)	0.04 *
The presence of belching	18.31 (15.5–21.01)	18.185 (17.218–21.3223)	>0.05	89.368 (64.809–143.23)	92.543 (69.999–163.337)	>0.05	47.999 (39.462–116.474)	56.608 (36.096–117.515)	>0.05
The presence of globus	46.961 (38.622–92.644)	18.06 (16.68–20.49)	0.04 *	88.58 (65.89–318.808)	93.738 (61.596–754.104)	>0.05	47.999 (40.157–63.894)	46.961 (32.027–350.4)	>0.05
The presence of odynophagia	16.68 (14.88–21.18)	18.95 (17.645–21.188)	>0.05	98.1 (66.879–318.808)	77.189 (64.284–107.214)	>0.05	53.74 (41.256–113.237)	46.961 (34.489–122.245)	>0.05
The presence of dysphagia	18.345 (15.75–21.41)	18.185 (16.252–20.228)	>0.05	92.541 (61.023–127.189)	90.358 (67.711–363.224)	>0.05	49.467 (39.263–78.487)	49.2545 (36.675–158.262)	>0.05
The presence of hoarseness	18.3 (16.18–21.18)	16.68 (12.3–25.665)	>0.05	91.347 (69.907–138.279)	93.738 (61.596–754.104)	>0.05	50.302 (39.065–113.237)	46.961 (32.027–350.4)	>0.05
The presence of post-nasal drip	17.29 (15.733–26.055)	18.38 (16.538–20.578)	>0.05	95.92 (57.363–179.548)	88.974 (70.184–148.276)	>0.05	46.656 (37.368–66.532)	51.467 (39.809–137.479)	>0.05
The presence of dyspnea	16.685 (15.733–21.8)	18.595 (16.555–20.925)	>0.05	93.538 (2.015–155.391)	89.171 (69.907–183.411)	>0.05	49.467 (39.61–90.871)	49.255 (38.186–137479)	>0.05
The presence of cough	18.045 (15.733–20.298)	16.68 (12.3–25.665)	>0.05	90.358 (69.175–148.276)	93.738 (61.596–754.104)	>0.05	51.467 (38.622–155.049)	46.941 (32.026–350.4)	>0.05

Abbreviation: IL-8, interleukin-8; TNF-α, *tumor necrosis factor*-α; IL-17, interleukin-17. *: variable with statistical significance.

**Table 4 jcm-13-05832-t004:** Correlation of IL-8, TNF-, and IL-17 serum levels with the symptom perception quantified by the total score of GERD-Q and FSSG, as well as the frequency of specific symptoms.

	Correlation Coefficient (ρ)	IL-8 (*p*)	Correlation Coefficient (ρ)	TNF-α (*p*)	Correlation Coefficient (ρ)	IL-17 (*p*)
Total scores of GERD-Q	0.148	>0.05	0.117	>0.05	0.184	>0.05
Total scores of FSSG	0.074	>0.05	0.029	>0.05	0.022	>0.05
The frequency of heartburn	0.094	>0.05	0.006	>0.05	0.273	0.01 *
The frequency of regurgitation	0.177	>0.05	0.349	0.04 *	0.129	>0.05
The frequency of epigastric pain	0.069	>0.05	0.095	>0.05	0.033	>0.05
The frequency of nausea	0.159	>0.05	0.042	>0.05	0.08	>0.05
The frequency of sleep disturbance	0.082	>0.05	0.029	>0.05	0.095	>0.05
The frequency of bloating	0.097	>0.05	0.022	>0.05	0.12	>0.05
The frequency of post-prandial fullness	0.020	>0.05	0.053	>0.05	0.036	>0.05
The frequency of early satiety	0.030	>0.05	0.005	>0.05	0.049	>0.05
The frequency of belching	0.159	>0.05	0.042	>0.05	0.080	>0.05
The frequency of globus	0.105	>0.05	0.032	>0.05	0.032	>0.05
The frequency of odynophagia	0.219	>0.05	0.080	>0.05	0.018	>0.05
The frequency of dysphagia (RSI 4)	0.044	>0.05	0.119	>0.05	0.049	>0.05
The frequency of hoarseness (RSI 1)	0.041	>0.05	0.063	>0.05	0.078	>0.05
The frequency of post-nasal drip (RSI3)	0.011	>0.05	0.063	>0.05	0.193	>0.05
The frequency of dyspnea (RSI 6)	0.100	>0.05	0.076	>0.05	0.032	>0.05
The frequency of cough (RSI7)	0.178	>0.05	0.072	>0.05	0.030	>0.05

*: variable with statistical significance.

## Data Availability

Data is unavailable due to privacy or ethical restrictions.

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
