# Peer review of "IL-8, TNF-α, and IL-17 in the Development of Erosive Esophagitis and Symptom Perception in Gastroesophageal Reflux Disease (GERD)"

_jcm, 2024, doi:10.3390/jcm13195832_

Round 1

Reviewer 1 Report

Comments and Suggestions for Authors

Dear Authors

Please correct the title Table 8. TNF-α, and IL-17 in the development of erosive esophagitis and symptoms perception in gastroesophageal reflux disease (GERD). Remove Table 8 from title.

In abstract, remove the space between lines 40 and 41.

Rewrite the conclusion of the abstract. You can not affirm that IL-8 serum levels are associated with mucosal damage. The correlation was with globus not with heartburn and regurgitation.

Methods

Why patients did not perform 24-hour pH monitoring? Globus is not a typical symptom of GERD and pH-monitoring is crucial to confirm or discharge GERD

The endoscopies were performed by the same endoscopist?

Why all patients were biopsied? 

Please separate the cytokines part of the methodology. Why were only these cytokines analyzed? Include the ELISA reading spectrum.

It is difficult to analyze cytokines in patients in this way, considering that several aspects can alter the level of cytokines. The ideal would be to evaluate the presence of these cytokines by immunohistochemistry.

A conclusão não reflete o teor do trabalho. Não há como correlacionar erosão com níveis séricos. Há necessidade de reformulação.

Comments on the Quality of English Language

Some errors need to be improved.

Author Response

  1. Abstract: remove the space between lines 40 and 41.

Rewrite the conclusion of the abstract. You cannot affirm that IL-8 serum levels are associated with mucosal damage. The correlation was with globus not with heartburn and regurgitation.

Answer: We have removed the space between lines 40 and 41.

We have revised the abstract to only indicate a possibility of correlation between IL-8 and mucosal damage. correlation

2    Methods

  • Why did patients not perform 24-hour pH monitoring? Globus is not a typical symptom of GERD and pH-monitoring is crucial to confirm or discharge GERD

Answer: we did not do pH monitoring study because of logistical and resources issue

  • The endoscopies were performed by the same endoscopist?

Answer: yes

  • Why all patients were biopsied? 

Answer: All patients were biopsied as per the protocol of the study, since we aimed to examine the relationship between cytokine levels, various symptoms, and histological findings

  • Please separate the cytokines part of the methodology. Why were only these cytokines analyzed? Include the ELISA reading spectrum.

Answer: We followed the reviewer advise and we separated the cytokine part of the methodology and elaborated on the ELISA measurement details. We chose these cytokines since there was plenty data available for other cytokines such as IL-1 and IL-6. Therefore, we focused on the IL-8, TNF alpha, and IL-17

  • It is difficult to analyze cytokines in patients in this way, considering that several aspects can alter the level of cytokines. The ideal would be to evaluate the presence of these cytokines by immunohistochemistry.

Answer:

Thank you for your comments. We realized this was one of the limitations to our study and tried to minimize biases by excluding subjects with acute illness, signs of infection, autoimmune disease, or malignancy that might have influenced the cytokines’ level. We realize the most ideal way to measure the cytokines is to measure the histological presence or the level of expression of these cytokines. However, such tests were very costly. We have further elaborated this limitation of our study in the discussion.

Reviewer 2 Report

Comments and Suggestions for Authors

1, It's a pleasure to review your manuscript. Thanks for your work.

2,The role of IL-8, TNF-α, and IL-17 in the development of erosive esophagitis and symptoms perception in GERD. This is an interesting topic.

3,The study evaluated the predictive significance of immune indexes by measuring the differences according to the severity of esophagitis.

4,The study design is sound and the data are reliable. However, it is well known that immune indicators(TNF-α, and IL-17) can be influenced by numerous factors. In the case of esophagitis, the results may be confounded by other inflammatory factors, leading to abnormal immune indicators.

5,Still eliminate potential factors in the design, but it's not sufficient. It is recommended to discuss other factors that might affect TNF-α and IL-17.

6,IL-8 is a cytokine that is significantly involved in the development of erosive esophagitis,

7,Can you provide the standard frequency or a specific value of presence in Table 三?

Author Response

Comments and Suggestions for Authors

1. It's a pleasure to review your manuscript. Thanks for your work.

  1. The role of IL-8, TNF-α, and IL-17 in the development of erosive esophagitis and symptoms perception in GERD. This is an interesting topic.

  1. The study evaluated the predictive significance of immune indexes by measuring the differences according to the severity of esophagitis.

  1. The study design is sound and the data are reliable. However, it is well known that immune indicators(TNF-α, and IL-17) can be influenced by numerous factors. In the case of esophagitis, the results may be confounded by other inflammatory factors, leading to abnormal immune indicators.

Answer:

We thank the reviewer for his/her comments.

We realized this was one of the limitations to our study and tried to minimize biases by excluding subjects with acute illness, signs of infection, autoimmune disease, or malignancy that might have influenced the cytokines’ level. We realize the most ideal way to measure the cytokines is to measure the histological presence or the level of expression of these cytokines. However, such tests were very costly. We have further elaborated this limitation of our study in the discussion

  1. Still eliminate potential factors in the design, but it's not sufficient. It is recommended to discuss other factors that might affect TNF-α and IL-17.

Issue Answer: We have addressed this in the limitation of the study section

  1. IL-8 is a cytokine that is significantly involved in the development of erosive esophagitis,

  1. Can you provide the standard frequency or a specific value of presence in Table 3?

We have realized that Table- 3 raised many questions and was unclear to the reviewer.  Therefore, we decided to split the table into two tables and added more description and specifications to each table